# A Meta-Analysis of Group Cognitive Behavioral Therapy and Group Psychoeducation for Treating Symptoms and Preventing Relapse in People Living with Bipolar Disorder

**DOI:** 10.3390/healthcare10112288

**Published:** 2022-11-15

**Authors:** Marcus K. Tan, Eik-Chao Chia, Wilson W. Tam, Roger S. McIntyre, Zhisong Zhang, Vu A. Dam, Tham T. Nguyen, Hoa T. Do, Roger C. Ho, Cyrus S. H. Ho

**Affiliations:** 1Yong Loo Lin School of Medicine, National University of Singapore, Singapore 117597, Singapore; 2Alice Lee School of Nursing, Yong Loo Lin School of Medicine, National University of Singapore, Singapore 117597, Singapore; 3Brain and Cognition Discovery Foundation, Toronto, ON M5S 1A1, Canada; 4Mood Disorders Psychopharmacology Unit, University Health Network, Toronto, ON M5S 1A1, Canada; 5Department of Pharmacology and Toxicology, University of Toronto, Toronto, ON M5S 1A1, Canada; 6Department of Psychiatry, University of Toronto, Toronto, ON M5S 1A1, Canada; 7Faculty of Education, Huaibei Normal University, Huaibei 235000, China; 8Institute for Global Health Innovations, Duy Tan University, Thanh Khê, Da Nang 550000, Vietnam; 9Faculty of Medicine, Duy Tan University, Thanh Khê, Da Nang 550000, Vietnam; 10Institute of Health Economics and Technology (iHEAT), Hanoi 100000, Vietnam; 11Institute for Health Innovation and Technology (iHealthtech), National University of Singapore, Singapore 119276, Singapore; 12Department of Psychological Medicine, Yong Loo Lin School of Medicine, National University of Singapore, Singapore 117597, Singapore; 13Department of Psychological Medicine, National University Health System, Singapore 119228, Singapore

**Keywords:** bipolar disorder, depression, psychotherapy, group, cognitive behavioral therapy, psychoeducation, maintenance, recurrence, mania, relapse, chronic disease

## Abstract

**Objectives:** This meta-analysis aims to evaluate the treatment outcomes of patients treated with Group cognitive behavioural therapy (GCBT) or group psychoeducation (GPE) as an adjunct to pharmacotherapy. **Methods:** Systematic search of PubMed, EMBASE, PsycINFO, and CENTRAL from inception till 1 March 2022 was conducted. Randomized-controlled-trials (RCTs) comparing GCBT/GPE with controls (treatment-as-usual/individualized therapy) in adults with bipolar disorder were eligible. The outcomes were relapse rates of any depressive or manic episodes and control of depressive and manic symptoms post-intervention. Overall odds-ratio was used to evaluate the relapse rates. Standard Mean Differences were pooled using a random-effects model for the control of depressive and manic symptoms. **Results:** 25 articles were assessed full-text independently by two members, and 11 studies were included in this meta-analysis. 601 and 590 participants were randomized into group-therapy (GCBT/GPE) and control, respectively. GPE significantly reduces relapse rates at post-intervention with Odds ratio of 0.43 (95% CI = 0.28-to-0.62, *p* < 0.0001) (I² = 41%) compared to control, however, no significant results were found for GPE on control of depressive or manic symptoms. No significant results were found for GCBT in all outcomes. **Conclusion:** This meta-analysis provides some evidence that GPE could be an efficacious treatment as an adjunct to treatment-as-usual in reducing the relapse rates of patients with bipolar disorder.

## 1. Introduction

Bipolar disorder, which is characterized by episodes of depressed, manic, and mixed mood states, is a lifelong debilitating mental disorder that is associated with morbidity and mortality [1]. With the global prevalence rate of bipolar disorder estimated at 2% and Disability-adjusted life years (DALYs) at 9.29 million years in 2017 [2,3]. It is evident that bipolar disorder has a significant impact on one’s cognition, function, and quality of life. In addition, bipolar disorder affects the quality of life of caregivers and family members, not to mention the burden on healthcare systems in terms of resources and costs around the world [4,5]. The impact of bipolar disorder is far-reaching. Bipolar Disorder is mainly treated with psychotropic medications including mood stabilizers and antipsychotic medications [6]. However, there is still a high relapse rate within two years after recovery [7]. Hence it is crucial to find an effective method in delaying and preventing relapse or recurrence in bipolar disorder.

Recent evidence suggests that adjunctive psychotherapies such as Cognitive Behavioral Therapy (CBT) or psychoeducation (PE) with psychotropic medications is more effective than medications alone in reducing overall relapse rates and recurrence of both depressive and manic episodes in people with bipolar disorder, improving an individual’s compliance and coping skills [8]. CBT, through cognitive restructuring, seeks to identify, evaluate, and modify dysfunctional thoughts during the individual’s session with a psychologist [9]. PE aims to provide patients with information about their conditions to enhance medication adherence and be able to detect early signs of mood changes. Psychotherapies are either conducted as individualized therapy or as a group therapy. Though individualized psychotherapy as an adjunct to the treatment of bipolar disorder have been studied and proven to be effective in reducing overall relapse rates based on the meta-analysis conducted by Chiang, K.J. et al. in 2017 for CBT and Rabelo, J.L. et al. for PE in 2021 [10,11].

Group Cognitive Behavioral Therapy (GCBT) seeks to apply the principles of cognitive behavioral therapy on a larger group of individuals. These include cognitive restructuring, use of exposure exercises and problem solving [9]. With group sizes ranging from 8 to 14 and the number of sessions from 10 to 20, GCBT helps individuals identify problems, recognize thoughts, and differentiate facts from irrational thoughts. Through those methods, GCBT challenges the dysfunctional assumptions present and allows individuals to learn and cope with these thoughts in their everyday lives. On the other hand, Group Psychoeducation (GPE) is a general term that refers to any intervention that educates the patients and their family members about the disease condition with the aim to improve overall outcomes. These can range from education on medication to enhance adherence to education on lifestyle management and coping strategies [12]. The group sizes usually vary from 5 to 15 in a group and the number of sessions varies depending on the objective of the programs in the respective institutions, and they generally stay within their allocated groups. As the therapist can see multiple patients at one session, the advantages of group therapy include cost-effectiveness and shorter waiting time. The disadvantages of group therapy include lack of privacy and negative group dynamics (e.g., splitting among group members) [13]. Beynon et al. (2008) and Bond K. et al. (2015) performed systematic review on effectiveness of GPE in treating bipolar disorder but lacked statistical analysis [8,12]. Currently, the National Institute for Health and Care Excellence (NICE) Guidelines recommends adjunctive psychotherapies as an adjunct to psychotropic medications with the adjunctive psychotherapies being cognitive behavioral therapy and psychoeducation, either as an individual or in a group [14]. However, the guidelines also emphasized that the results have a low quality of evidence and are not statistically significant in terms of outcome measures. As a result, the effectiveness of group therapy, including GPE or GCBT, as an adjunct treatment for bipolar disorder remains debatable and inconclusive [12,15].

Therefore, this study aims to evaluate the effectiveness of GCBT or GPE in the treatment of people with bipolar disorder as an adjunct therapy to psychotropic medications. A meta-analysis of RCTs was performed to evaluate the efficacy of GCBT and GPE on the relapse rates, depressive and manic symptoms post-intervention.

## 2. Methods

This meta-analysis followed the methodology recommended by the Cochrane Handbook for Systematic Reviews of Interventions and reported based on the Preferred Reporting for Items for Systematic Reviews and Meta-Analyses (PRISMA) guidelines [16,17].

### 2.1. Inclusion Criteria

The following criteria was employed: (i) study population: adults above the age of 18 formally diagnosed with bipolar disorder, either Bipolar I or II disorder; (2) intervention: GCBT or GPE as an adjunct therapy to the psychotropic medications; (3) comparator: controls, such as peer support group or individualized therapy or treatment-as-usual (TAU), were eligible; (4) outcomes: the outcomes were that of any relapses that occur post-intervention and control of depressive or mania symptoms post-intervention. The total number of randomized participants, post-intervention or follow-up mean and standard deviation (SD)/95% confidence interval (CI) in one or more of the stated outcomes should be reported in eligible studies. Change from baseline means and standard deviations would not be used in this analysis as it introduces an additional source of bias by which the baseline and final outcome measure will be reported for different numbers of participants due to loss-to-follow-up and study withdrawal [17]. The depressive symptoms were assessed using validated tools such as the Hamilton Rating Scale for Depression (HRSD), the Beck Depression Inventory (BDI), the Beck Hopelessness Scale (BHS), or the Montgomery–Asberg Depression Rating Scale (MADRS); the severity of manic symptoms was assessed with validated tools such as the Young Mania Rating Scale (YMRS) or Mania Rating Scale (MRS); and (5) design of study: randomized controlled trial.

### 2.2. Exclusion Criteria

The exclusion criteria employed was: (1) studies that are written in non-English, (2) ongoing trials or studies or abstracts from conferences, (3) cross-over studies or cluster RCTs, and (4) studies including individuals below 18 years old, people with alcohol or substance abuse or comorbid Axis I disorders or participants with neurological conditions.

### 2.3. Database Search

Databases including PubMed, PsychINFO, EMBASE, and CENTRAL were searched systemically from inception until 1 March 2022. For the database search, Boolean Operators and synonyms were employed. For example, the search strategy for GCBT in PubMed was as follows: (Bipolar disorder OR bipolar OR manic-depressive psychosis OR bipolar affective disorder OR bipolar depression) AND (Group Cognitive therapy OR Group behavioral therapy OR Group Cognitive Behavioral Therapy OR Group CBT OR Group Psychotherapy). Appendix A provides the full search strategies employed in the database search. The reference sections of prior meta-analysis on CBT, GCBT and GPE were also searched manually.

### 2.4. Study Selection

After the duplicate studies were removed, titles and abstracts of all remaining search results were screened independently by 2 reviewers for eligibility in this study. Articles that met the inclusion criteria were assessed independently in full text by the two reviewers. A third reviewer would be consulted in the event of disagreements.

### 2.5. Data Extraction

Extraction of data were performed by the first author with the second author performing cross checks of the studies. From each study, the following data were sought: (1) Characteristics of the study, such as the initial author, year, country, study arms (Intervention versus control), population characteristics, attrition rates of GCBT or GPE and controls (the number of randomized participants absent at follow-up or evaluation after intervention was completed), and GCBT or GPE rates of non-completion (the number of GCBT participants who were unable to complete all the intervention sessions as stated in the study); (2) intervention details such as GCBT duration, components, and control condition; (3) baseline study demographics such as the number of randomized participants in the intervention and control groups, number of females, mean age of each group, and (4) outcome data, such as the number of participants who completed the study, post-intervention or follow-up means and SD/95% CI. When further clarification is needed, the corresponding authors of published trials were contacted for further information.

### 2.6. Bias Assessment

The revised Cochrane risk of bias tool for randomized trials was used in this study to evaluate the risk of bias in the included studies [18]. Five domains of risk of bias were assessed: (1) randomization method, (2) deviations from the intended interventions, (3) missing data outcome, (4) outcome measurement and (5) selective reporting. The bias risk was evaluated and was given one of the three categories as stated: ‘low risk of bias’, ‘some concerns’, or ‘high risk of bias’.

### 2.7. Statistical Analysis

The efficacy of GCBT or GPE in lowering the relapse rate was evaluated from the overall odds ratio (OR). Standardized Mean difference (SMD) was used as the effect measure for the efficacy of GCBT or GPE in the control of depressive or manic symptoms. Pooled SMD between the groups were calculated by comparing the differences between GCBT or GPE and the control group in post-intervention outcome measure. The SMD was used as the effect measure for both depressive and mania symptoms as these are reported using different rating scales in various RCTs, this allows the effect outcome to be interpreted and compared across various RCTs in this meta-analysis [19]. Due to the clinical diversity and differences across the studies, the DerSimonian and Laird’s random effects model was used [20]. This model makes the assumption that the included studies used in this meta-analysis are samples that are picked at random from a wider population. This seeks to extrapolate the findings and conclusions from the selected studies used [21]. Heterogeneity between the studies were assessed via I^2^ statistics. Publication bias was assessed with funnel plots and Egger’s regression. RevMan 5.4. was used for statistical analysis and Egger’s regression was performed using the “metafor” function in R version 4.1.0 software.

## 3. Results

### 3.1. Study Selection and Study Characteristics

The PRISMA diagram (Figure 1) reports the study selection process conducted in this study. The number of unique articles identified were 1380, from which, 25 full text articles were accessed for eligibility. 11 articles of 11 different studies were included and analyzed in this study. The reasons for the exclusion of the 14 articles were stated in the flow-diagram.

Table 1 summarizes the characteristics of the 11 studies included in this meta-analysis. 601 and 590 participants were randomized into group therapy (GCBT or GPE) and control conditions respectively. Of which, for GCBT, 120 and 109 participants were randomized into GCBT and control conditions, respectively, while for GPE, 481 and 481 participants were randomized into GPE and control conditions, respectively. The mean age of the participants enrolled was from 33 to 45 years. Out of the 11 RCTs, four studies used GCBT while 7 used GPE. The number of participants per session of group therapy ranged from 6 to 14 and 6 to 18 for GCBT and GPE, respectively. The number of intervention sessions was from 12 to 18 and 8 to 21 for GCBT and GPE, respectively. Other characteristics of the included studies are found in Table 1 while excluded studies can be found in Appendix A.

Most of the 11 studies used in this meta-analysis were evaluated to be of low bias risk in the randomization method (90.9%), missing data outcome (100%), selective reporting (72.7%), and low risk of bias in the outcome measurement (63.6%) (Appendix A). 54.5% of the 11 studies used were rated of some concerns due to potential risk of bias for deviations from intended intervention and lack of blinding by those delivering the intervention.

### 3.2. Outcome (1)—Relapse Rates

In the 9 RCTs (Figure 2) reporting on the treatment outcomes involving the relapse rates, the pooled effect for GCBT does not show any favorable response in terms of reduction in overall relapse rates for the participants under the GCBT group over the control condition, (OR = 0.72, 95% CI = 0.19 to 2,66, Z = 0.50, *p* = 0.62) (I² = 67%, *p* = 0.08). However, the pooled effect for GPE showed a significant favorable response in terms of reduction in relapse rates of GPE as opposed to the control condition (OR = 0.43, 95% CI = 0.28 to 0.62, Z = − 4.14, *p* < 0.0001) (I² = 41%, *p*= 0.12). This finding suggested that GPE demonstrated a significant impact in reducing the relapse rates of people with bipolar disorder.

### 3.3. Outcome (2)—Severity of Depression and Mania

In the 5 RCTs (Figure 3) reporting on the treatment outcomes involving severity of depression, which employed BDI, MADRS, and HDRS-17, the pooled effective size for GCBT and GPE, respectively, did not indicate any significant favorable response in terms of changes in depressive symptoms as compared to the control condition. For GCBT, the SMD between GCBT and the control is −0.11 (95% CI = −0.77 to 0.55, Z = 0.33, *p* = 0.74) (I² = 77%, *p* = 0.01) while for GPE, the SMD between GPE and the control is −0.07 (95% CI = −0.26, 0.11, Z = 0.76, *p* = 0.45) (I² = 0%, *p* = 0.66).

In the 5 RCTs (Figure 4) reporting on the treatment outcomes involving severity of manic symptoms, which were assessed by the YMRS and Bech-Rafaelsen Mania Scale, the pooled effect size for GCBT and GPE do not indicate any significant reduction in manic symptoms as compared to control condition. For GCBT, the SMD between GCBT and the control condition is 0.24 (95% CI = −0.08 to 0.56, Z = 1.48, *p* = 0.14) (I² = 10%, *p* = 0.33) while for GPE, the SMD between GPE and the control is −0.04 (95% CI = −0.23, 0.14, Z = 0.44, *p* = 0.66) (I² = 0%, *p* = 0.42).

### 3.4. Publication Bias

The funnel plots (Appendix A for the post-intervention relapse rates, severity of depressive and manic symptoms appear symmetrical. In the funnel plot for depressive symptoms, there is an outlier whereby that study has a higher standard error as opposed to the other studies. Egger’s Regression test was calculated for relapse rates (−2.02, 95% CI = −4.99 to 0.94, *p* = 0.15), depressive (−0.61, 95% CI = −7.45 to 6.24, *p* = 0.79) and manic (0.82, 95% CI = −4.28 to 5.92, *p* = 0.64) symptoms indicating no publication bias.

## 4. Discussion

Overall, this meta-analysis provides evidence that GPE may be an efficacious treatment as an adjunct to pharmacological treatments in reducing the relapse rates of people with bipolar disorder. Our results are in concordance with previous systematic reviews. Earlier analyses, however, had relatively lesser RCTs available for analysis, Bond K et al. 2015 (*n* = 5, overlap = 3) and Beynon S. et al. 2008 (*n* = 2) [8,12]. We postulate that GPE may reinforce medication adherence and thereby reduce the risk of relapse.

With regards to the severity of post-intervention depressive and manic symptoms, this meta-analysis found that GPE did not reduce severity of symptoms after intervention. Our finding is not in accordance with Bond K et al. (2015) who suggested that GPE demonstrated evidence of reduction in post-intervention symptoms of depression and mania. For GCBT, there were no significant differences in post-intervention relapse rates, severity of both post-intervention depressive and manic symptoms between the intervention and control groups. To our knowledge, no other meta-analysis covering GCBT has been conducted before.

There are a few postulated reasons as to having a more significant effect in reduction in relapse rates in GPE compared to GCBT. Firstly, as GPE has shown to improve reported medication adherence and treatment literacy of participants [12]. A potential difficulty in conducting GCBT is the need to cater to the varying cognitive styles of individuals. Hence, strategies formulated and taught during the sessions may not be suited for every individual present, potentially leading to poor adherence to the coping strategies caught during the session [33]. Secondly, people with bipolar disorder may find GPE less stigmatizing as it offers education to the whole group and is less likely to discuss sensitive areas (e.g., hypersexuality during manic phase) as the case for GCBT in front of the whole group. This suggests that GPE requires fewer sessions as compared to GCBT to achieve its goals, in a study conducted by Parikh, S. V. 2012, patients were randomized to undergo either 6 sessions of GPE or 20 sessions of Individual CBT, and it was found that both had similar outcomes in terms of their reduction in symptom burden and relapse rates [34]. Finally, the effects of GCBT are only seen during the therapy sessions and up to 6 months after the sessions while the effects attenuated afterwards [8,33]. However, the number of studies that were selected and included in our meta-analysis was small, coupled with the small sample sizes of these studies, this might prevent us from finding any significant differences between the post-intervention outcomes of GCBT and control.

There are a few limitations of this study. The studies included in GCBT have small sample sizes, less than 100 [22,23,24,25] and it might not achieve the adequate statistical power to detect the differences between GCBT and control conditions. Further studies with a larger sample size would be required to examine the effects of GCBT more accurately on the post-intervention relapse rates and control of depressive and manic symptoms. In addition, the relapse rates calculated and used in this meta-analysis were calculated based on the number of relapses that occurred during the follow-up of the patient’s post-intervention. The follow-up periods used in the studies varied and ranged from 1 to 6 years, which would likely have affected the number of relapses that occurred during the follow-up period. The definition of relapses or recurrence of a depressive or manic episode also differed among studies because some studies defined relapse or recurrence based on scoring higher than a cut-off score in an outcome measure (HDRS-17, YMRS), while some studies defined relapse or recurrence as hospitalization due to a depressive or manic episode. As a result, those studies that equated hospitalization rates as relapse rates might under-report the number of relapses because some depressive or manic episodes might not require hospitalization. Lastly, the outcome measures used to measure post-intervention depressive or manic symptoms were measured by different questionnaires including HDRS-17 (*n* = 3), BDI (*n* = 1), MADRS (*n* = 1), YMRS (*n* = 4) and BRMS (n=1). Finally, this study protocol was not registered as part of PROSPERO or any other register of systemic reviews, which may lead to outcome and reporting bias in this review.

## 5. Conclusions

This meta-analysis provides evidence that GPE could be an efficacious treatment as an adjunct to TAU in reducing relapse rates of people with bipolar disorder. For GCBT, while current evidence may not be compelling with respect to reducing overall relapse rates and recurrence of manic and depressive episodes, further studies should be conducted analyzing the potential use of GCBT for other therapeutic targets as well. CBT via virtual platforms should be analyzed as an adjunct in the treatment of bipolar disorders.

## Figures and Tables

**Figure 1 healthcare-10-02288-f001:**
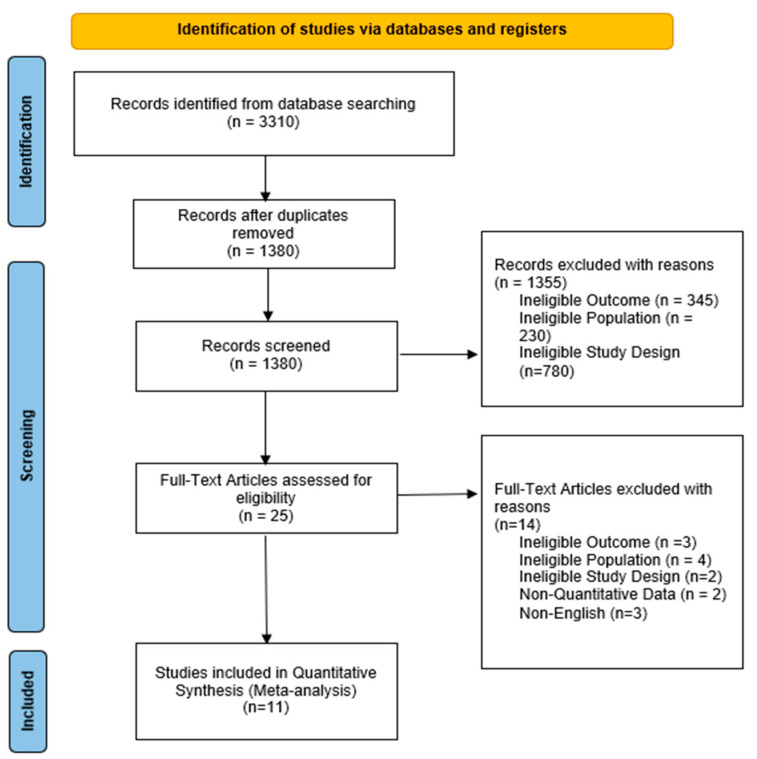
PRISMA Flowchart for Study Selection.

**Figure 2 healthcare-10-02288-f002:**
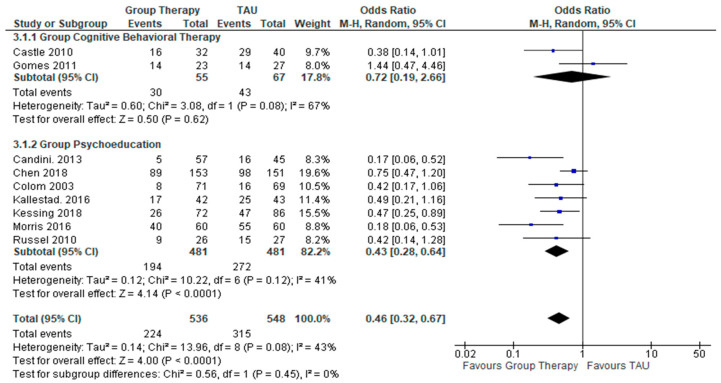
Forest plots for relapse rates after interventions.

**Figure 3 healthcare-10-02288-f003:**
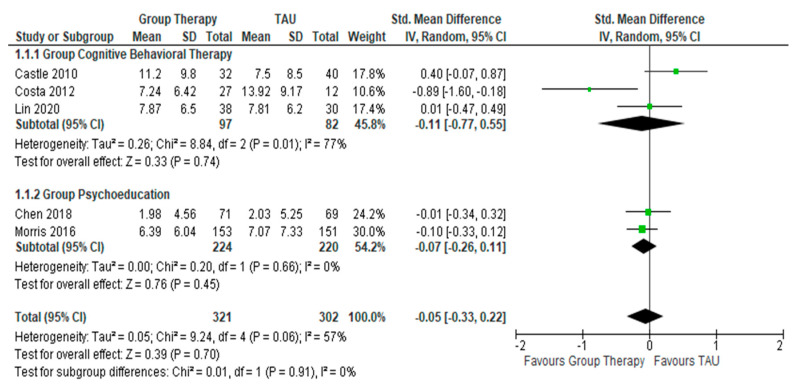
Forest plots for the changes in severity of depressive symptoms after interventions.

**Figure 4 healthcare-10-02288-f004:**
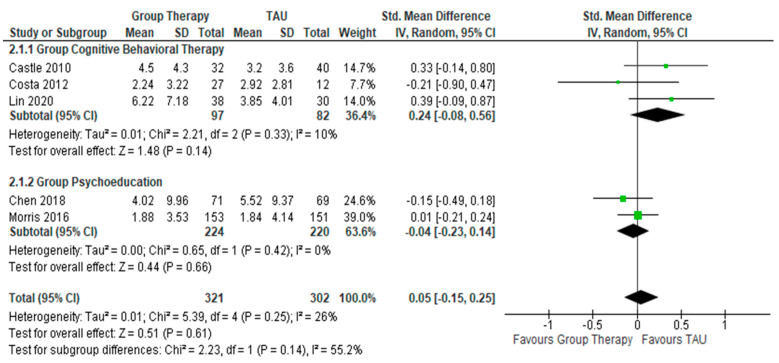
Forest Plots for the changes in severity of manic symptoms after interventions.

**Table 1 healthcare-10-02288-t001:** Characteristics of Included Studies.

Study	Country	Bipolar Definition	Study Design	Sample Demographics (Patients)	Intervention Characteristics	Outcomes
**Group Cognitive Behavioral Therapy (GCBT)** Total: 229						
Castle et al. 2018 [22]	Australia	DSM-IV-TR	2 parallel arms (Group CBT with TAU, TAU)	Age 18–65 years Total, *n*: 72 Mean Age: 42 Gender (M/F): 20/52	No. of sessions: 12 + 3 booster Group Size: 10–14	Relapse Rates Depression (MADRS) Mania (YMRS) Assessed Time Point: 12 months
Costa et al. 2012 [23]	Brazil	DSM-IV	2 parallel arms (Group CBT with TAU, TAU)	Age 18–60 years Total, *n*: 39 Mean Age: 41 Gender(M/F): 21/18	No. of sessions: 14 Group Size: N.A.	Depression (BDI) Mania (YMRS) Assessed Time points: Before, During and After treatment and up to 6 month follow-up
Gomes et al. 2011 [24]	Brazil	DSM-IV	2 parallel arms (Group CBT with TAU, TAU)	Age 18–60 years Total, *n*: 50 Mean Age: 34 Gender (M/F): 12/38	No. of sessions: 18 Group Size: >6	Relapse Rates Assessed Time Points: 12 months
Lin et al. 2020 [25]	Taiwan	DSM-IV	2 parallel arms (Group CBT with TAU, TAU)	Age > 18 years Total, *n*: 68 Mean Age: 41 Gender (M/F): 26/42	No. of sessions: 12 + 3 booster Group Size: 8–12	Depression (HDRS-17) Mania (YMRS) Assessed Time Point: Every 3 months for 1 year
**Group Psychoeducation (GPE)** Total: 962						
Candini et al. 2013 [26]	Italy	DSM-IV	2 parallel arms (Group Psychoeducation with TAU, TAU)	Age 18–65 years Total, *n*: 102 Mean Age: 43 Gender (M/F): 50/52	No. of sessions: 21 Group Size: 8–12	Relapse Rate Assessed Time Points: 1 year
Chen et al. 2018 [27]	China	DSM-V	2 parallel arms (Group Psychoeducation with TAU, TAU + Regular Free discussions)	Age 18–60 years Total, *n*: 140 Mean Age: 33 Gender (M/F): 14/126	No. of sessions: 8 Group Size: 8–12	Relapse Rates Depression (HDRS-17) Mania (YMRS) Assessed Time Points: 12 Months
Colom et al. 2003 [28]	Spain	DSM-IV	2 parallel arms (Group Psychoeducation + TAU, TAU)	Age 18–65 years Total, *n*: 120 Mean Age: N.A. Gender (M/F): 44/76	No. of sessions: 21 Group Size: 8–12	Relapse Rate Assessed Time Points: 2 years
Kallestad et al. 2016 [29]	Norway	DSM-IV	2 parallel arms (Group Psychoeducation + TAU, Individual Psychoeducation + TAU)	Age > 18 years Total, *n*: 85 Mean Age: 38 Gender (M/F): 39/46	No. of sessions: 10 + 8 Booster Group Size: 8–12	Relapse Rates Assessed Time Points: 27 months
Kessing 2018 [30]	Denmark	ICD-10	2 parallel arms (Group Psychoeducation + TAU, TAU)	Age 18–70 years Total, *n*: 158 Mean Age: 36 Gender (M/F): 72/86	No. of sessions: 12 + 3 booster Group Size: 6–8	Relapse Rate Assessed Time Points: 6 years
Morris 2016 [31]	United Kingdom	DSM-IV	2 parallel arms (Group Psychoeducation/TAU + Unstructured Peer Support)	Age > 18 years Total, *n*: 304 Mean Age: 45 Gender (M/F): 127/177	No. of sessions: 21 Group Size: 10–18	Depression (HDRS-17) Mania (BRMS) Relapse Rates Assessed Time points: Weekly for 96 weeks
Russel 2010 [32]	Australia	DSM-IV	2 parallel arms (Group Psychoeducation + TAU, TAU)	Age > 18 years Total, *n*: 53 Mean Age: 40 Gender (M/F): 28/30	No. of sessions: 12 Group Size: N.A.	Relapse Rate Assessed Time Points: 60 weeks

DSM: Diagnostic and Statistical Manual of Mental Disorders, ICD: International Classification of Diseases and Related Health Problems, TAU: Treatment-as-usual, N.A.: Not applicable or Not explicitly stated, MADRS: Montgomery–Åsberg Depression Rating Scale, YMRS: Young Mania Rating Scale, BDI: Beck’s Depression Inventory, HDRS-17: Hamilton Depression Rating Scale3.2. Risk of Bias.

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
