# Peer review of "A Meta-Analysis of Group Cognitive Behavioral Therapy and Group Psychoeducation for Treating Symptoms and Preventing Relapse in People Living with Bipolar Disorder"

_healthcare, 2022, doi:10.3390/healthcare10112288_

Round 1

Reviewer 1 Report

This paper is well designed and clearly described. It also provides clinically useful contents.

However, it would be better to briefly introduce the detailed composition of the GCBT and GPE analyzed in this study.

Reviewer 2 Report

The manuscript is clearly, well structured and written. Methodology is also good.  I miss if there are an age interval or it is >18y to ....

My concern is in relation to the references and the evidences for the use of the two therapies reviewed. I miss some relevant as the reference to clinical guides.  

Reviewer 3 Report

This is a meta-analysis aimed at evaluating treatment outcomes of patients treated with Group cognitive behavioural therapy (GCBT) or group psychoeducation (GPE) as an adjunct to pharmacotherapy in bipolar disorder. The authors found that GPE could be an efficacious treatment as an adjunct to treatment as usual in the clinical care of patients with bipolar disorder. The methodology appears sound but I have some questions:

-       It is not clear why cross over studies or cluster RCTs were excluded

-       Sub analysis according to the diagnostic subgroup should be performed

-       The search string appears limited in its definition

Reviewer 4 Report

I would like to thank you for your valuable study. This study focused on group therapy, Group Cognitive Behavioral Therapy (GCBT) and Group Psychoeducation (GPE) and conducted a meta-analysis to discuss their effects. In the absence of systematic discussion about the therapeutic effects of GCBT, I think that this study is great because it is extremely challenging.

However, this study has serious weaknesses in its methodology. I, therefore, consider it difficult to publish. This does not mean denying the value of this study. I hope this study will be improved to make it better.

Major revisions:

1.       Analysis designs

1.1. Small numbers of studies in the meta-analysis

The main concern is the small number of studies. When the meta-analysis was conducted separately by the outcome, there were only two or three studies, except for the GPE with the relapse rates as the outcome. Also, each study has a small sample size. The small sample size may explain why the therapeutic effects of GCBT and some of the therapeutic effects of GPE were not statistically significant. I consider that discussing the therapeutic effects of GCBT and GPE from these results may cause great misunderstanding among readers.

Specifically, although you mention the small number of studies and sample size in the Discussion section, you argued in the next paragraph (“There are a few postulated reasons….”) that GPE is effective and GCBT is ineffective. A nonsignificant result does not mean no effect. Therefore, such considerations are not desirable. If you want to discuss the results of the meta-analysis, you should interpret them from the effect size and the range of its confidence intervals rather than discussing whether they are significant or not. 

The following are my suggestions for improvements. I hope this helps you.

1.       I would recommend systematic reviews over meta-analyses as the methodology for this study. I don't recommend trying to adjust statistically for the small number of studies. There are 2 ~ 3 studies each, so it seems possible to compare and discuss each study. It would be easier for readers to understand the therapeutic effects of GCBT and GPE if the 11 studies were systematically presented.

2.       It might be better to moderate the selection rule to ensure a larger number of studies. For example, conducting a meta-analysis could include non-English articles or non-RCT studies. Or, if you don't focus on Bipolar Disorder, you could target more symptoms.

 1.2. unit of analyses

GCBT and GPE may have treatment effects that differ not only among individuals but also among treatment groups. You should need multilevel analyses?

2.       Definitions of GCBT and GPE

There was a lack of explanation about GCBT and GPE in the Introduction section.

What are these definitions? What kind of structure did the treatment groups have? Is homogeneity of the treatment group important? How many did therapists participate in the therapies? Is there any common guideline? Did the members change from section to section?

Also, why did you select only studies with participants over the age of 18? Is it related to the features of group therapy?

3.       Table 1

Table 1 was explained on p.5. It is unclear, however, how the values on p.5 were calculated from the values in Table 1. For example, the Total N for the GCBT studies in Table 1 sums to 225, but you mention that the GCBT and control conditions for the GCBT studies were 120 and 109 (total 229), which do not match.

Please add to Table 1 the sample size for each condition.  

4.       Minor revision

Please revise the following three points.

4.1. p.2 Introduction

×(e.g. splitting among group members [12]

(e.g. splitting among group members) [12]

4.2. p.2 Introduction

×systematic review on effectiveness of GPE in treating bipolar disorder

systematic review on effectiveness of group psychoeducation (GPE) in treating bipolar disorder

4.3. p.2 Introduction

×the effectiveness of group therapy including group psychoeducation (GPE) or group CBT

the effectiveness of group therapy including GPE or Group Cognitive Behavioral Therapy (GCBT)

Round 2

Reviewer 4 Report

Thank you for responding to my comments. 

I am still reluctant to vehemently argue for the present results obtained with a small number of studies and a small sample size. I think that the fact that Cochrane's guidelines permit does not guarantee statistical validity. However, the authors have improved on this point by improving expression by revising the sentence structure. I also understand that a multilevel analysis is beyond the scope of this paper.

Also, thank you for adding the definitions of GCBT and GPE. It made the discussions throughout the paper much easier to understand.

Thank you also for the correction of Table 1. I would like to request the following correction in the text on page 5.

× while for GPE, four hundred eight one and 481 participants were …

while for GPE, 481 and 481 participants were …
